# Effects of a Mixed *Limosilactobacillus fermentum* Formulation with Claimed Probiotic Properties on Cardiometabolic Variables, Biomarkers of Inflammation and Oxidative Stress in Male Rats Fed a High-Fat Diet

**DOI:** 10.3390/foods10092202

**Published:** 2021-09-17

**Authors:** Micaelle Oliveira de Luna Freire, Luciana Caroline Paulino do Nascimento, Kataryne Árabe Rimá de Oliveira, Alisson Macário de Oliveira, Thiago Henrique Napoleão, Marcos dos Santos Lima, Cláudia Jacques Lagranha, Evandro Leite de Souza, José Luiz de Brito Alves

**Affiliations:** 1Biotechnology Center, Department of Biotechnology, Federal University of Paraíba, João Pessoa 58051900, PB, Brazil; mica_macario@hotmail.com; 2Health Sciences Center, Department of Nutrition, Federal University of Paraiba, João Pessoa 58051900, PB, Brazil; lucianacpnascimento@hotmail.com (L.C.P.d.N.); katarynearabe_@hotmail.com (K.Á.R.d.O.); evandroleitesouza@hotmail.com (E.L.d.S.); 3Biological Sciences Center, Department of Biochemistry, Federal University of Pernambuco, Recife 50670901, PE, Brazil; alissonmacario@hotmail.com (A.M.d.O.); thiago.napoleao@ufpe.br (T.H.N.); 4Department of Food Technology, Federal Institute of Sertão Pernambucano, Petrolina 56302100, PE, Brazil; marcos.santos@ifsertao-pe.edu.br; 5Laboratory of Biochemistry and Exercise Biochemistry, Federal University of Pernambuco, Vitória de Santo Antão 55608680, PE, Brazil; lagranha@hotmail.com

**Keywords:** high-fat diet, inflammation, oxidative stress, probiotic, *Limosilactobacillus* *fermentum*

## Abstract

High-fat diet (HFD) consumption has been linked to dyslipidemia, low-grade inflammation and oxidative stress. This study investigated the effects of a mixed formulation with *Limosilactobacillus*
*fermentum* 139, *L.* *fermentum* 263 and *L.* *fermentum* 296 on cardiometabolic parameters, fecal short-chain fatty acid (SCFA) contents and biomarkers of inflammation and oxidative stress in colon and heart tissues of male rats fed an HFD. Male Wistar rats were grouped into control diet (CTL, n = 6), HFD (n = 6) and HFD with *L.* *fermentum* formulation (HFD-Lf, n = 6) groups. The *L.*
*fermentum* formulation (1 × 10^9^ CFU/mL of each strain) was administered twice a day for 4 weeks. After a 4-week follow-up, biochemical parameters, fecal SCFA, cytokines and oxidative stress variables were evaluated. HFD consumption caused hyperlipidemia, hyperglycemia, low-grade inflammation, reduced fecal acetate and propionate contents and increased biomarkers of oxidative stress in colon and heart tissues when compared to the CTL group. Rats receiving the *L*. *fermentum* formulation had reduced hyperlipidemia and hyperglycemia, but similar SCFA contents in comparison with the HFD group (*p* < 0.05). Rats receiving the *L.* *fermentum* formulation had increased antioxidant capacity throughout the colon and heart tissues when compared with the control group. Administration of a mixed *L.* *fermentum* formulation prevented hyperlipidemia, inflammation and oxidative stress in colon and heart tissues induced by HFD consumption.

## 1. Introduction

Impairment in gut microbiota composition, gut dysbiosis and enhanced systemic inflammation have been reported in cardiometabolic disorders, such as obesity, diabetes, stroke hypercholesterolemia and heart failure [1,2], suggesting that alterations in the “gut–heart axis” could be involved in pathogenesis of cardiometabolic disorders.

Excessive high-fat diet (HFD) consumption can trigger gut dysbiosis, a state characterized by impairment of gut microbiota diversity and increased intestinal permeability [3,4]. In addition, an HFD is likely to promote oxidative stress in the colon [5], low-grade chronic inflammation [6], cardiac oxidative stress, ventricular dysfunction [7], increased blood pressure, autonomic dysfunction and metabolic disorder [8]. Altogether, these findings suggest an association involving the gut–heart axis in HFD-induced cardiometabolic disorders [9].

Gut microbiota modulation through probiotic use has received special attention as a safe approach for the prevention and/or treatment of cardiometabolic dysfunction [10]. Probiotics have been defined as live microorganisms that confer health benefits to the host when administered in adequate doses [11]. *Lactobacillus* and amended genera are the most common genera of probiotics, being commonly recognized as safe and with qualified presumption of safety. The genus *Lactobacillus* was recently reclassified into 25 genera [12]. In the new proposed taxonomic reclassification, *Lactobacillus fermentum* was renamed as *Limosilactobacillus fermentum* and described as Gram-positive, rod- or coccoid-shaped, heterofermentative and anaerobic or aerotolerant, being found in fermented cereals and other fermented plant materials, dairy products, manure, sewage and the feces and vagina of humans [12].

It has been demonstrated that probiotics, when administered as a single strain or mixed strains, can exert health-promoting effects on the host through different mechanisms, such as the normalization of unbalanced gut microbiota, production of short-chain fatty acids (SCFAs), increased turnover of enterocytes, colonization resistance and competitive exclusion of pathogens [11]. Additionally, some probiotics have displayed strain-specific effects, such as: production of specific bioactive metabolites, enhanced activity of the immune system at the intestinal or extraintestinal level [13] and enhanced activity of antioxidant enzymes, which cause elimination of reactive oxygen species in the host intestine and alleviation of oxidative damage [14]. Thus, the identification of strain-specific qualities or mechanisms in potentially probiotic microorganisms should be relevant and required in the development and applicability of a probiotic product.

In recent years, our research group has isolated and characterized potentially probiotic fruit-derived strains. The strains of *L. fermentum* 139, *L. fermentum* 296 and *L. fermentum* 263 were recovered from Brazilian fruit by-products [15,16]. *L. fermentum* 139 was isolated from *Mangifera indica* L. (mango), *L. fermentum* 263 was isolated from *Ananas comosus* (pineapple) and *L. fermentum* 296 was isolated from *Fragaria vesca* L. (strawberry). All the three strains displayed potential for use as probiotics in terms of a set of functionality-related in vitro properties, such as auto-aggregation, co-aggregation, survival during exposure to simulated gastrointestinal conditions and pathogen antagonism, in addition to showing the absence of hemolytic and mucolytic activities and resistance to antibiotics [16]. Such findings indicated that these *L. fermentum* fruit-derived strains could be potential candidates for use as novel probiotics.

Early investigations of our laboratory verified that administration of *L. fermentum* 296 alone [8] or a mix with three *L. fermentum* strains [17] reduced blood pressure, autonomic dysfunction and dyslipidemia in rats. However, the effects of *L. fermentum* administration on immune and enzymatic activities, which are strain-specific functional characteristics and not reported in all probiotic strains, remain to be elucidated. Here, we have evaluated the effects of a mixed formulation with three potentially probiotic *L. fermentum* strains on cardiometabolic parameters, fecal SCFA contents and biomarkers of inflammation and oxidative stress in colon and heart tissues of rats fed an HFD. 

## 2. Methods

### 2.1. Animals and Ethical Aspects

Male Wistar rats (*Rattus norvegicus*, 100 days of age) were used in this study. The rats were kept in collective polypropylene cages (3 animals/cage) with controlled temperature (22 ± 1 °C), humidity (50–55%) and light–dark cycle (12 h), receiving water and diet ad libitum. The procedures were in accordance with the National Council for Control of Animal Experimentation (CONCEA), and International Principles for Biomedical Research. The experimental protocols were approved by an Institutional Animal Care Committee (CEUA-UFPB protocol number # 6080240418).

### 2.2. Probiotic Strains and Preparation of Probiotic Suspension 

The strains of *L. fermentum* 139, *L. fermentum* 263 and *L. fermentum* 296 were provided by the Laboratory of Food Microbiology, Department of Nutrition, Federal University of Paraíba (João Pessoa, Brazil). Stocks were kept at −20 °C in de Mann, Rogosa and Sharpe (MRS) broth (HiMedia, Mumbai, India) with glycerol (Sigma-Aldrich, St. Louis, MO, USA; 20 mL/100 mL). Suspensions of probiotic cells were prepared from overnight cultures grown in MRS broth under anaerobiosis (Anaerobic System Anaerogen, Oxoid Ltda., Wade Road, UK) at 37 °C [8,17]. The mixed cell suspension (counts of approximately 9 log CFU/mL of each strain) was prepared with the mixture of the suspension of each strain (ratio 1:1:1). 

### 2.3. Experimental Design

Rats were grouped into: (i) a control group (CTL, n = 6), fed with a control diet prepared according to the American Institute of Nutrition—AIN-93M [18]; (ii) an HFD group, fed with a high-fat diet (HFD, n = 6) purchased from Rhoster^®^ Company (Araçoiaba da Serra, São Paulo, Brazil) and receiving a placebo; and (iii) an HFD group receiving the formulation with *L. fermentum* 139, *L. fermentum* 263 and *L. fermentum* 296 (HFD-Lf, n = 6). Compositions of the CTL diet and HFD are shown in Table 1. 

Phosphate-buffered saline (PBS) was given as a placebo for 4 weeks in the CTL and HFD groups. The *L. fermentum* formulation in a PBS solution of approximately 3 × 10^9^ CFU/mL was administered twice a day for 4 weeks to the HFD-Lf group. Administration of placebo or *L. fermentum* formulation was carried out with oral gavage. Body weight was measured every 3 days during the experimental period with an appropriate scale (model AS-1000; Marte, Santa Rita MG, Brazil). After 4 weeks, rats were euthanized by decapitation and biochemical parameters and cytokines were measured in serum; acetic and propionic acids were measured in feces; and oxidative stress parameters were measured in colon and heart tissues.

### 2.4. Quantification of Organic Acids in Colonic Contents

Acetic, butyric and propionic acids were measured with a high-performance liquid chromatography (HPLC) technique using an LC 1260 Infinity system (Agilent Technologies, St. Clara, CA, USA) coupled to a photo diode detector array (PDA) detector (G1315D; Agilent Technologies) as previously described [19].

### 2.5. Biochemical Analysis and Atherogenic Indices

Measurements of levels of total cholesterol, high-density lipoprotein cholesterol (HDL-c), triglycerides and glucose in serum were taken using commercial kits and a HumaLyzer 3500 semi-automatic photometer (HUMAN Gesellschaft für Biochemica und Diagnostica mbH, Wiesbaden, Germany). Levels of low-density lipoprotein cholesterol (LDL-c) were calculated with the Friedewald equation: LDL-c (mg/dL) = [TC − HDL-c − TG]/5 [20].

Atherogenic indices were calculated as follows: cardiac risk ratio (CRR) = total cholesterol/HDL-c [21]; atherogenic index of plasma (AIP) = triglycerides/HDL-c [22] and Castelli’s risk index II (CRI-II) = LDL-c/HDL-c [23].

Levels of cytokines (IL-6, IL-10, IL1β and TNF-α) were measured with a Millipore 7-plex kit (Millipore Corp., Billerica, MA, USA) following the manufacturer’s instructions. The estimation of the levels of cytokines was carried out from a standard curve using a third order polynomial equation, expressed as pg/mL. Samples with levels of cytokines below the limit of detection were recorded as zero, while samples with levels of cytokines above the upper limit of quantification of standard curves were assigned the highest value of the curve.

### 2.6. Measurement of Oxidative Stress in Colon and Heart

Heart and colon tissues were homogenized using a cold buffer solution (50 mM Tris and 1 mM EDTA (pH 7.4), 1 mM sodium orthogonadate and 200 μg/mL of phenylmethanesulfonyl fluoride) with an IKA RW 20 digital homogenizer, a Potter–Elvehjem pestle and glass tubes on ice. Homogenates were centrifuged (1.180 g, 10 min, 4 °C) [24] and levels of proteins were determined using the Bradford protocol [25]. The homogenates of heart and colon tissues (0.3 mg/mL) were used to measure the lipid peroxidation, enzymatic activities and total thiol contents.

Lipid peroxidation was quantified by the production of malondialdehyde (MDA) in reaction with thiobarbituric acid (TBA) at 100 °C. Sequential additions of trichloroacetic acid and Tris-HCl (3 mM) were carried out, followed by centrifugation (2.500 g, 10 min, 0.8% (*v*/*v*)). Afterwards, TBA was added to the resultant supernatant, mixed and boiled for 15 min. After cooling, the reaction was read at 535 nm on a spectrophotometer.

Enzymatic activity of total superoxide dismutase (SOD) was determined following the Misra and Fridovich method. Tissue homogenates were incubated with sodium carbonate buffer (0.05% (*p*/*v*), pH 10.2, 0.1 mmol/L EDTA) at 37 °C. Next, 30 mM/L of epinephrine (in 0.05% acetic acid) was added and SOD activity was measured by the kinetics of epinephrine auto-oxidation inhibition for 1.5 min at 480 nm [26].

Catalase activity was measured by decomposition of H_2_O_2_ into O_2_ and H_2_O. Tissue homogenates were incubated with 50 mM phosphate buffer (pH 7.0). Next, 0.3 M of H_2_O_2_ was added and absorbance was read at 240 nm for 1.5 min [27].

For measurement of glutathione S-transferase (GST) activity [28], tissue homogenates were added to phosphate buffer (0.1 M, pH 6.5 with 1 mM EDTA), 1 mM 1-chloro-2,4-dinitrobenzene (CDNB) and 1 mM of reduced glutathione (GSH). Absorbance was read at 340 nm for 1.5 min.

For measurement of total thiol groups, tissue homogenates were incubated (30 min) in phosphate buffer with 10 mM of 5,5′-dithiobis (2-nitrobenzoic acid) in the dark. The absorbance was read at 412 nm [29].

### 2.7. Statistical Analysis

The results were described as mean ± standard deviation for parametric data or median (maximum–minimum) for non-parametric data. A Kolmogorov–Smirnov test was used to assess data normality. Parametric variables were analyzed with one-way ANOVA and a Tukey post hoc test. Non-parametric variables were compared with a Kruskal–Wallis test with Dunn’s post hoc test. A Pearson’s or Spearman correlation coefficient (r) was used to evaluate the relationships among biochemical and inflammatory parameters and SCFA contents. The correlations were classified as bad (r ≤ 0.20), weak (0.21–0.40), moderate (0.41–0.60), good (0.61–0.80) and excellent (0.81–1.00). Statistical analysis was carried out with Prism 6 software (GraphPad Software 6, San Diego, CA, USA). A *p*-value of <0.05 was considered significant.

## 3. Results

### 3.1. Body Weight, Biochemical Parameters and Cytokine Serum Levels

The percentage of weight gain at the end of the protocol was similar among groups (Table 2). Rats fed an HFD had higher serum levels of glucose, total cholesterol, LDL-c and triglycerides, as well as higher atherogenic indices in comparison with the CTL group (Table 2). Additionally, rats fed an HFD had higher serum levels of proinflammatory cytokines TNF-α and IL-1β and lower serum levels of IL-6 and IL-10 in comparison with the CTL group (Table 2). Administration of a mixed *L. fermentum* formulation effectively reduced serum levels of glucose, triglycerides, total cholesterol, LDL-c, proinflammatory cytokine IL1β and atherogenic indices, as well as increased serum levels of HDL-c and anti-inflammatory cytokine IL-10 in rats fed an HFD (Table 2). These results show that supplementation of a mixed *L. fermentum* formulation had ameliorative effects on dyslipidemia, atherogenic indices and low-grade inflammation in rats fed an HFD.

### 3.2. Short-Chain Fatty Acids, Fructose and Raffinose in Feces

HFD consumption decreased fecal acetic (0.05 ± 0.02 vs. 0.17 ± 0.11 g/L, *p* < 0.05) and propionic acid contents (0.30 ± 0.17 vs. 0.81 ± 0.23 g/L, *p* < 0.05) in comparison with the CTL group (Figure 1A,B). Butyric acid contents were below the limit of detection. Administration of a mixed *L. fermentum* formulation did not change fecal contents of SCFAs (Figure 1A,B).

Fecal propionic acid contents correlated negatively with serum levels of cholesterol (r = −0.61, *p* = 0.006), LDL-c (r = −0.67, *p* = 0.002) and triglycerides (r = −0.67, *p* = 0.002), but did not correlate with HDL-c serum levels (r = 0.13, *p* = 0.60) (Figure 2A–D). Fecal propionic acid contents correlated negatively with serum levels of TNF-α (r = −0.67, *p* = 0.002, Figure 2E) and IL-1β (r = −0.72, *p* = 0.0008, Figure 2F), and correlated positively with serum levels of IL-6 (r = 0.65, *p* = 0.03, Figure 2G), and IL-10 (r = 0.61, *p* = 0.007, Figure 2H).

### 3.3. Oxidative Stress Biomarkers in Colon Tissues

Rats fed an HFD had increased MDA levels (1.2 ± 0.4 vs. 0.6 ± 0.1 nmol/mg protein, *p* < 0.05), and decreased SOD (346.2 ± 29.8 vs. 417.2 ± 15.1 U/mg protein, *p* < 0.05) and CAT activities (1.9 ± 0.5 vs. 5.0 ± 1.7 U/mg protein, *p* < 0.05) in colon tissues in comparison with the CTL group (Figure 3A–C). GST activity and sulfhydryl contents in colon tissues were similar between CTL and HFD groups (*p* > 0.05, Figure 3D,E). In comparison to the HFD group, supplementation of a mixed *L. fermentum* formulation, despite not showing improved CAT activity (*p* > 0.05), reduced the MDA levels (0.5 ± 0.2 vs. 1.2 ± 0.4 nmol/mg protein, *p* < 0.05) and increased SOD (435 ± 46 vs. 346.2 ± 29.8 U/mg protein, *p* < 0.05) and GST activities (22.0 ± 6.4 vs. 12.7 ± 3.6 U/mg protein, *p* < 0.05) in colon tissues (Figure 3A–D). In addition, administration of a mixed *L. fermentum* formulation increased the sulfhydryl content in the colon in comparison with HFD and CTL groups (*p* < 0.05, Figure 3E).

### 3.4. Oxidative Stress Biomarkers in Heart Tissues

Levels of MDA and SOD activity in heart tissues were similar among groups (*p* > 0.05, Figure 4A,B). Rats fed an HFD had reduced CAT (4.1 ± 0.4 vs. 8.1 ± 2.5 U/mg protein, *p* < 0.05) and GST activities (24.3 ± 6.7 vs. 33.9 ± 2.2 U/mg protein, *p* < 0.05) in heart tissues in comparison with the CTL group (Figure 4C,D). Total sulfhydryl content was decreased in heart tissue of the HFD group in comparison with the CTL group (0.13 ± 0.02 vs. 0.19 ± 0.03 mmol/mg protein, *p* < 0.05, Figure 4E). Administration of a mixed *L. fermentum* formulation restored CAT (7.3 ± 0.8 vs. 4.1 ± 0.4 U/mg protein, *p* < 0.05) and GST activities (36.9 ± 5.9 vs. 24.3 ± 6.7 U/mg protein, *p* < 0.05), as well as sulfhydryl content (0.21 ± 0.04 vs. 0.13 ± 0.02 mmol/mg protein, *p* < 0.05) in heart tissues in comparison with the HFD group (Figure 4C–E).

## 4. Discussion

The results of this study showed that administration of a mixed formulation with *L. fermentum* 139, *L. fermentum* 263 and *L. fermentum* 296 twice a day for 4 weeks caused improvements in cardiometabolic parameters, and decreased systemic low-grade inflammation and biomarkers of oxidative stress in colon and heart tissues induced by excessive HFD consumption in rats.

It was previously demonstrated that administration of a single strain, *L. fermentum* 296, for 4 weeks reduced cholesterol and triglyceride serum levels in rats fed an HFD, but did not increase the HDL-c levels, and did not improve glucose tolerance and insulin sensitivity [8]. In this study, the administration of a mixed *L. fermentum* formulation effectively decreased the serum levels of glucose, cholesterol, LDL-c, triglycerides and atherogenic indices, and increased HDL-c serum levels, demonstrating superior hypolipidemic and hypoglycemic effects in rats fed an HFD. Similarly, early investigations have demonstrated that administration of *L. fermentum* FTDC 8312, *L. fermentum* MJM60397, *L. fermentum* ME-3 and *L. fermentum* MTCC: 5898 had hypolipidemic effects in rodents [30,31] and humans [32].

Some lactobacilli and amended genera strains may bind to cholesterol in the intestine, causing increased fecal cholesterol excretion. In addition, colonic bacteria may produce SCFAs from carbohydrate or protein fermentation, which has been linked to inhibition of hepatic and gut cholesterol synthesis, an energy source for colonocytes, maintenance of gut barrier integrity and anti-inflammatory effects [10,33,34]. Acetic, propionic and butyric acids are the most predominant SCFAs in the colon, accounting for 90–95% of total colonic SCFAs [35]. In the present study, the administration of a mixed *L. fermentum* formulation did not increase SCFA fecal contents in rats fed an HFD. However, fecal propionic acid contents were correlated negatively with serum levels of cholesterol, LDL-c, triglycerides, TNF-α and IL-1β, and correlated positively with IL-10 and IL-6, indicating that dyslipidemia and pro-inflammatory conditions should be associated with reduced fecal propionic acid contents. These results are in accordance with early findings demonstrating that fecal and circulating propionate could exert inhibitory effects on “de novo” lipogenesis and cholesterogenesis in the liver [36,37]. Colonic propionate may also inhibit NF-κB signaling and reduce expression of pro-inflammatory factors, such as TNF-α and IL-1β in colon tissues [38], as well as induce IL10-producing Treg cells [39], which are related as a novel candidate to improve metabolism and inflammation [40].

The gut microbiota is greatly responsive to dietary modifications, with it being verified that HFD consumption can cause gut dysbiosis through oxidative stress in gut mucosa, decrease the population of gut barrier-protecting bacteria and increase the population of endotoxin-producing bacteria [41,42]. The impairment in the gut antioxidant system provoked by HFD consumption can lead to disruption of intestinal epithelial tight junctions, damage to intestinal mucosal integrity, translocation of indigenous intestinal bacteria, systemic low-grade inflammation through the activation of nuclear factor kappa B (NF-κB) pathways and upregulation of TNF-α and IL-1, with increased risk of peripheral organ damage and systemic metabolic dysfunction [43,44,45]. The results of this study have demonstrated that HFD consumption for 4 weeks reduced the activities of antioxidant enzymes and increased the MDA levels in colon tissues, as well as induced low-grade inflammation and cardiometabolic disorders, in rats.

Oxidative stress tolerance and antioxidant capacity have been reported in probiotic strains [14,46]. Early studies have reported antioxidant effects in several *L. fermentum* strains, such as *L. fermentum* MTCC: 5898 [31], *L. fermentum* ME-3 [47], *L. fermentum* CQPC07 [46], *L. fermentum* CECT5716 [48] and *L. fermentum* I5007 [49]. Here, we have demonstrated that administration of a mixed formulation with *L. fermentum* 139, *L. fermentum* 263 and *L. fermentum* 296 effectively reduced MDA levels and increased SOD and GST activity and sulfhydryl content (for non-enzymatic antioxidant defense) in colonic mucosa of rats fed an HFD, indicating an antioxidant capacity of the tested strains.

The mechanism involved in the antioxidant capacity of some probiotic strains is still under investigation. However, it has been suggested that antioxidant capacity of *L. fermentum* strains could be associated with a fully functional GSH system formed by GSH peroxidase and GSH reductase, which acts to protect cells against oxidative stress [14]. In fact, this information corroborates the results of our study, which have shown an increase in total thiol content. The improvement in antioxidant capacity in colonic mucosa promoted by administration of a mixed *L. fermentum* formulation was associated with a downregulation in systemic low-grade inflammation, and alleviation of oxidative stress in heart tissues of rats fed an HFD.

An early study demonstrated that daily supplementation of *L. fermentum* KBL374 and *L. fermentum* KBL375 (1 × 10^9^ CFU, for 8 days) alleviated inflammation in the gut through regulation of immune responses, and alteration of gut microbiota in a dextran sulfate sodium-induced colitis model [50]. Similarly, daily administration of *L. fermentum* MTCC: 5898 (2 × 10^9^ CFU, for 12 weeks) reduced gene expression of cytokines TNF-α and IL-6 in the liver of rats fed a cholesterol-enriched diet [31].

Pro- and anti-inflammatory properties, as well as regulation of metabolic, regenerative and neural processes have, been reported as potential functions of IL-6 [51]. Administration of a mixed *L. fermentum* formulation did not change the IL-6 serum level, but reduced the IL-1β level and upregulated the serum level of anti-inflammatory cytokine IL-10 in rats fed an HFD. These results suggest that the *L. fermentum* formulation may alleviate low-grade inflammation provoked by HFD consumption.

The IL-10 is a cytokine exerting anti-inflammatory properties with a key role for the limitation of host immune responses to pathogens, which could prevent impairment in the host, and maintaining tissue homeostasis [52]. It has been reported that *L. fermentum* CECT5716 can increase regulatory T cells (Treg1) and IL-10-producing cells [53]. In addition, probiotic strains can reduce inflammation via downregulation of NF-κB pathways in in vitro and in vivo conditions [54]. The underlying mechanism involved in immune modulation caused by the examined *L. fermentum* formulation remains to be elucidated.

Previously, we demonstrated that HFD consumption for 4 weeks caused dysautonomia, cardiac baroreflex control impairment and increased blood pressure in male rats [8]. Systemic inflammation and a high level of oxidative stress can induce autonomic dysfunction [55,56], and some evidence has suggested that cardiac impairment may be linked to gut–heart axis damage [57,58]. The lack of gut microbiota composition analysis could be described as a main limitation of this study, although we have previously documented enhanced *Lactobacillus* counts in feces from rats treated with *L. fermentum* strains with claimed probiotic properties [8,17]. Another possible limitation of this study could be the lack of a well-established probiotic strain as a comparative reference. In fact, this was not conducted in earlier studies due to: (i) initial metabolic characterization of a potentially probiotic strain and (ii) different *Lactobacillus* (or amended genera) species analyzed. For example, regarding *L. fermentum* strains, there are several strains from different countries and isolated from diverse sources exhibiting antioxidant and anti-inflammatory properties. However, there has been no consensus on the strain of *L. fermentum* that should be used as a comparative reference for other studies. Indeed, this is an important aspect to be solved in the coming years.

In this study, we have shown for the first time that administration of a mixed *L. fermentum* formulation concomitantly increased GST activity and sulfhydryl content in colonic mucosa and heart tissue of rats fed an HFD, indicating that probiotic administration may impact directly on heart metabolism, probably via the gut–heart axis.

## 5. Conclusions

Administration of a formulation containing a mix of *L. fermentum* 139, *L. fermentum* 263 and *L. fermentum* 296 with claimed probiotic properties twice a day, for 4 weeks, caused hypocholesterolemia and hypoglycemic effects, increased HDL-c serum levels and alleviated loss of fecal SCFAs induced by HFD consumption. In addition, this study has shown that administration of the *L. fermentum* formulation may effectively decrease low-grade inflammation and biomarkers of oxidative stress in colon and heart tissues in rats fed an HFD. Finally, it may be reasonable to suggest that the examined *L. fermentum* formulation has great potential to act as a novel antidyslipidemia product due to its ability to attenuate lipid metabolism disorders, inflammation and oxidative stress.

## Figures and Tables

**Figure 1 foods-10-02202-f001:**
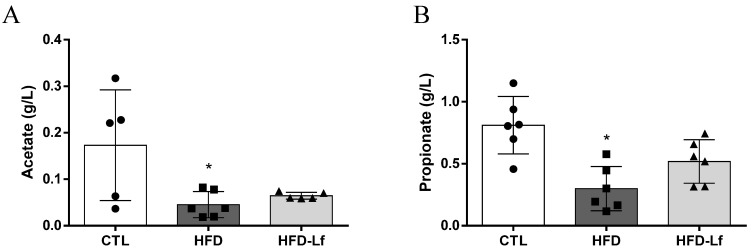
Effects of a mixed formulation with *L. fermentum* 139, *L. fermentum* 263 and *L. fermentum* 269 on short-chain fatty acid concentration in fecal samples of rats fed an HFD. Assessment of acetic (**A**) and propionic acids (**B**) in fecal samples. Groups: control (CTL, n = 6), high-fat diet (HFD, n = 6) and HFD with *L. fermentum* formulation (HFD-Lf, n = 6). Data are displayed as mean ± standard deviation, and were analyzed by ANOVA one-way test with Tukey post hoc test. * *p* < 0.05 indicates significant difference between HFD and CTL groups. During HPLC experiments, acetate contents of 2 rats (01 of CTL group and 01 of HFD-Lf group) were below the analytical detection limit.

**Figure 2 foods-10-02202-f002:**
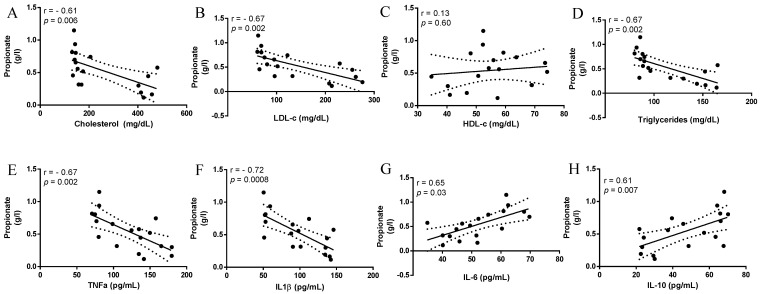
Assessment of correlation coefficients between propionic acid concentration, biochemical and cytokine variables. High-density lipoprotein cholesterol (HDL-c); low-density lipoprotein cholesterol (LDL-c) levels; tumor necrosis factor-α (TNF-α); interleukin 1-beta (IL1β); interleukin 6 (IL-6); interleukin 10 (IL-10).

**Figure 3 foods-10-02202-f003:**
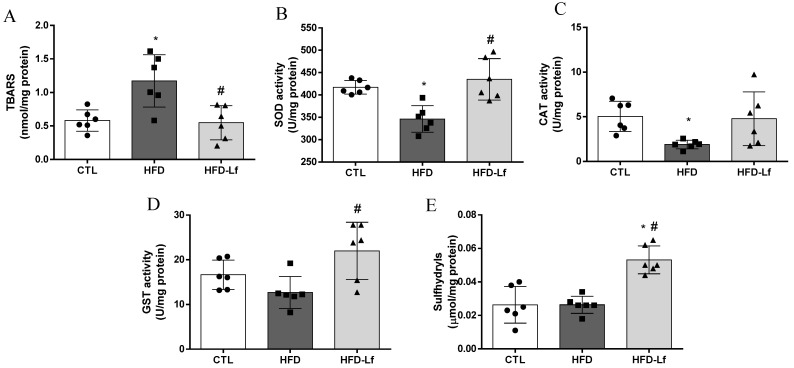
Effects of a mixed formulation with *L. fermentum* 139, *L. fermentum* 263 and *L. fermentum* 269 on oxidative stress parameters in colon mucosa of rats fed an HFD. Measurement of malondialdehyde levels (MDA, **A**), superoxide dismutase activity (SOD, **B**), catalase activity (CAT, **C**), glutathione S-transferase activity (GST, **D**) and total sulfhydryl content (**E**) in colon mucosa. Groups: control (CTL, n = 6), high-fat diet (HFD, n = 6) and HFD with *L. fermentum* formulation (HFD-Lf, n = 6). Data are displayed as mean ± standard deviation, and were analyzed by ANOVA one-way test with Tukey post hoc test. * *p* < 0.05 indicates significant difference between HFD or HFD-Lf and CTL groups; # *p* < 0.05 indicates significant difference between HFD-Lf and HFD groups.

**Figure 4 foods-10-02202-f004:**
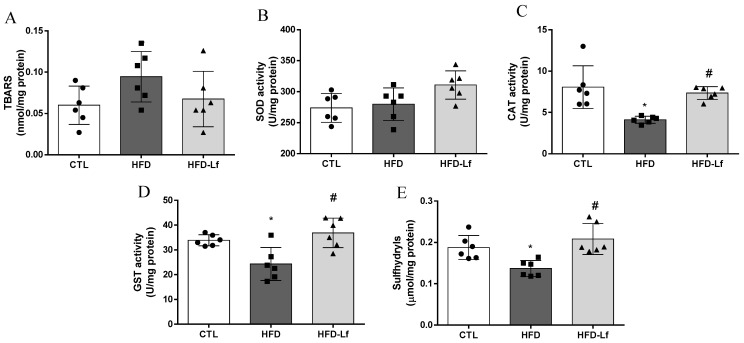
Effects of a mixed formulation with *L. fermentum* 139, *L. fermentum* 263 and *L. fermentum* 269 on oxidative stress parameters in heart tissue of rats fed an HFD. Measurement of levels of malondialdehyde (MDA, **A**), superoxide dismutase activity (SOD, **B**), catalase activity (CAT, **C**), glutathione S-transferase activity (GST, **D**) and total sulfhydryl content (**E**) in heart tissue. Groups: control (CTL, n = 6), high-fat diet (HFD, n = 6) and HFD with *L. fermentum* formulation (HFD-Lf, n = 6). Data are displayed as mean ± standard deviation, and were analyzed by ANOVA one-way test with Tukey post hoc test. * *p* < 0.05 indicates significant difference between HFD or HFD-Lf and CTL groups; # *p* < 0.05 indicates significant difference between HFD-Lf and HFD groups.

**Table 1 foods-10-02202-t001:** Composition of control and high-fat diet (HFD) offered.

Ingredients (g/100 g)	Diets	
	Control (AIN-93M) *	HFD **
Corn starch	39.75	33.09
Dextrinized corn starch	13.20	15.50
Casein ^#^	20.00	19.86
Sucrose	10.00	6.00
Soybean oil	7.00	3.00
Animal fat (lard)	0.00-	6.00
Non-hydrolyzed vegetable fat	0.00	5.00
Sigma cholesterol	0.00	1.00
Sigma colic acid	0.00	0.50
Cellulose	5.00	5.00
Mineral mix 93M	3.50	3.50
Vitamin mix	1.00	1.00
L-cystine	0.30	0.30
Choline bitartrate	0.25	0.25
t-BHQ ***	0.014	0.014
Nutritional composition		
Calories (Kj/100 g)	16.46	18.05
Carbohydrate (%)	63.8	50.5
Protein (%)	20.3	18.3
Lipids (%)	15.9	31.2

* Adapted from Reeves et al. (1993). ** Rhoster—Industry and Trade Ltd. *** t-BHQ: tert-Butylhydroquinone. ^#^ Casein showed 85% purity (85 g protein for each 100 g casein).

**Table 2 foods-10-02202-t002:** Body weight, serum levels of biochemical parameters, atherogenic indices and cytokines in rats fed a control (CTL), high-fat diet with a placebo (HFD) and HFD with a mixed formulation with *L. fermentum* 139, *L. fermentum* 263 and *L. fermentum* 269 (HFD-Lf) twice a day for 4 weeks.

	CTL (n = 6)	HFD (n = 6)	HFD-Lf (n = 6)	F	*p*-Value
**% Weight gain**	12.1 ± 2.9	6.6 ± 3.6	7.6 ± 4.4	2.9	0.09
**Biochemical parameters**					
Glucose (mmol/L)	5.8 ± 0.5	11.2 ± 2.0 *	8.2 ± 0.9 * ^#^	25.45	<0.0001
Triglycerides (mmol/L)	0.9 ± 0.03	1.6 ± 0.12 *	1.1 ± 0.06 * ^#^	69.05	<0.0001
Total cholesterol (mmol/L)	36.1 ± 1.4	113.5 ± 7.6 *	43.6 ± 5.6 * ^#^	360.3	<0.0001
LDL-cholesterol (mmol/L)	17.4 ± 1.4	62.5 ± 7.3 *	27.3 ± 5.0 * ^#^	128.4	<0.0001
HDL-cholesterol (mmol/L)	14.5 ± 1.0	10.6 ± 2.7 *	17.5 ± 1.6 * ^#^	17.47	0.0001
**Atherogenic indices**					
CRR	2.5 ± 0.2	11.3 ± 2.9 *	2.5 ± 0.5 ^#^	53.11	<0.0001
AIP	1.5 ± 0.1	3.5 ± 0.7 *	1.4 ± 0.2 ^#^	54.86	<0.0001
CRI-II	1.2 ± 0.2	6.2 ± 1.5 *	1.6 ± 0.3 ^#^	63.93	<0.0001
**Cytokine levels**					
TNF-α (pg/mL) †	78.4 (70.7–80.6)	139.3 (134.8–179.7) *	137.2 (98.8–158.0) *	11.79	0.0003
IL-6 (pg/mL)	63.8 ± 3.9	43.3 ± 5.8 *	49.3 ± 5.4 *	25.65	<0.0001
IL-1β (pg/mL)	52.6 ± 1.7	139.3 ± 4.4 *	96.7 ± 9.2 * ^#^	313.0	<0.0001
IL-10 (pg/mL)	66.4 ± 2.5	25.2 ± 3.6 *	49.4 ± 11.4 * ^#^	51.23	<0.0001

CRR: cardiac risk ratio. AIP: atherogenic index of plasma. CRI-II: Castelli’s risk index II. (*) indicates significant difference (*p* < 0.05) in comparison with CTL. (^#^) indicates significant difference (*p* < 0.05) in comparison with HFHC group. † Non-parametric data.

## Data Availability

The data that support the findings of this study are available on request from the corresponding author. The data are not publicly available due to privacy or ethical restrictions.

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
