# Peer review of "Effects of a Mixed Limosilactobacillus fermentum Formulation with Claimed Probiotic Properties on Cardiometabolic Variables, Biomarkers of Inflammation and Oxidative Stress in Male Rats Fed a High-Fat Diet"

_foods, 2021, doi:10.3390/foods10092202_

Round 1
Reviewer 1 Report
In this manuscript, the authors described the mixed L. fermen-tum formulation can prevent hyperlipidemia, inflammation and oxidative stress along gut-heart axis induced by HFD consumption. Overall, the manuscript is well written with a clear logic to follow. However, the manuscript falls short in providing essential proofs of the claims. Authors should compare HFD-LF with the reference to report properly and to discuss the results.
Author Response
In this manuscript, the authors described the mixed L. fermentum formulation can prevent hyperlipidemia, inflammation and oxidative stress along gut-heart axis induced by HFD consumption. Overall, the manuscript is well written with a clear logic to follow. However, the manuscript falls short in providing essential proofs of the claims. Authors should compare HFD-LF with the reference to report properly and to discuss the results.
We would like to thank the reviewer 1 for recognizing the relevance of our work. We appreciate the review of our manuscript and the great contributions that have been made. In fact, the problem highlighted by reviewer 1 is a potential limitation in most studies that analyze the properties of a new and potentially probiotic strains. In revised version, we have added this potential limitation of study, as follows:
“Another potential limitation of study was the lack of a well-established probiotic strain as a comparative reference. In fact, this has not been conducted by various studies due: i) initial metabolic characterization of a potentially probiotic strain; and ii) different Lactobacillus species analyzed. For example, regarding L. fermentum strains, there is several species from different countries and isolated from diverse sources exhibiting antioxidant and anti-inflammatory properties. However, there is still no consensus on the strain of L. fermentum that should be used as a comparative reference for studies. In the next years, this will need to be solved and international bank of well-cataloged probiotic strains with well-defined studies and information about the qualification of the deposited strains to a specific use could help to solve the problem”.
Reviewer 2 Report
The authors evaluate the effects of a mixed formulation containing Lactobacillus fermentum on cardiometabolic parameters, inflammatory markers, fecal short-chain fatty acid (SCFA) contents and oxidative stress in rats fed a HFD diet. Research in gut-heart axis field is important, however, there have been a lot of literatures confirming that administration of L. fermentum can improve cardiovascular disease. This study is not innovative or important. Also, about the mechanism of action of lactic acid bacteria on cardio-health, the parameters analyzed by the authors have also been discussed in other references. Although the author emphasizes the term of gut-heart axis in the manuscript, the analysis data has not been verified in this, which weakens the novelty of this article. Below are some points need to clarify or improve.
Major points:
- The authors should provide some evidence about fermentum treatment improves cardiovascular disease via gut-heart axis.
- Gut microbiota should be analyzed to show the correlation between gut-heart axis and L. fermentum.
- The authors should explain that why a mixed formulation with three L. fermentum strains were used in this study, rather than single strain or combined different species of Lactobacillus. Is there any data illustrate the effect of this three strains combined use is stronger than single strain?
- The author conducts animal experiments, but the data lacks tissue sections of intestinal or heart tissue, such as lesions, inflammation, and other related measurements.
Minor points:
- In Table 2, IL-1β and IL-6 belong to the inflammatory cytokine under the same inflammation pathway, and their levels in serum often show similar trends between groups. Since the administration of L. fermentum can significantly reduce the increase in IL-1β production induced by HFD, why this result is not seen in TNF-α and IL-6?
- In Figure 1, since n = 6 in each group, why only showed 5 spots in some groups?
- In Figure 2, the authors should explain why fecal propionate contents were correlated negatively with serum levels of cholesterol, LDL-c, and triglycerides but did not correlate with HDL-c serum levels. Also, fecal propionate contents were correlated negatively with serum levels of TNF-α/IL-1β and correlated positively with serum levels of IL-6/IL-10 should be discussed.
- The sentence “ fermentum … was administered two twice a day ...” should be replaced to ““L. fermentum … was administered twice a day ...”” in Abstract section.
- The terms of “ºC” in the Materials and Methods section should not underlined and should be replaced to “oC”.
- The terms of “CFU/Ml” in the Materials and Methods section should be replaced to “CFU/mL”.
- In the Materials and Methods section, “Homogenates were centrifuged at 1.180 g for 10 min”. Centrifugal speed should have type errors and should be corrected.
- In conclusion section, “…twice a day for four weeks” should be replaced to “…twice a week for four weeks”.
- When mention to bacteria, many species names are not italicized throughout the manuscript, please correct them.
Author Response
The authors evaluate the effects of a mixed formulation containing Lactobacillus fermentum on cardiometabolic parameters, inflammatory markers, fecal short-chain fatty acid (SCFA) contents and oxidative stress in rats fed a HFD diet. Research in gut-heart axis field is important, however, there have been a lot of literatures confirming that administration of L. fermentum can improve cardiovascular disease. This study is not innovative or important. Also, about the mechanism of action of lactic acid bacteria on cardio-health, the parameters analyzed by the authors have also been discussed in other references. Although the author emphasizes the term of gut-heart axis in the manuscript, the analysis data has not been verified in this, which weakens the novelty of this article. Below are some points need to clarify or improve.
We would like to thank the reviewer 2 for all considerations on our manuscript. We would like to highlight that, although early studies have demonstrated positive effects of L. fermentum administration on cardiovascular disease, the strains of L. fermentum investigated in early studies were not of Brazilian origin and neither fruit-derived. In our knowledge, this is the first study with strains originating of Brazil with great antioxidant, immuno-regulatory and lipid-lowering capacity with great capacity for a translational approach within a clinical study in the future. Why would this be important? Because all marked probiotics in Brazil are currently imported, generating a high cost for consumers of probiotics in our country. In addition, each country has a legislation on probiotics and not necessarily an isolated strain marketed in China, for example, could be easily imported and marketed in Brazil. In addition, there is a trend in several countries to develop a well-catalogued bank of probiotics properly identified metagenomically and functionally. Certainly, this identification is important in development of probiotic-products and their applicability in disease condition, since several effects are specific strains. In this sense, we do not agree with reviewer 2 that the study is not innovative or important.
Major points:
- The authors should provide some evidence about fermentum treatment improves cardiovascular disease via gut-heart axis.
Answer: We would like to thank the reviewer 2 for this suggestion. we have used the terms: lactobacillus fermentum and gut-heart axis; and limosilactobacillus fermenutm and gut-heart axis. Unfortunately, we didn't find results in pubmed.
- Gut microbiota should be analyzed to show the correlation between gut-heart axis and L. fermentum.
Answer: in fact, this is a limitation, but does not diminish the novelty of study, mainly because all procedeurs to isolation and characterization of probiotic strain was performed and published (DOI: 10.1007/s12602-017-9318-2) and early studies in vivo found increased lactobacillus counts and decreased enterobacteriaceae in feces (Doi: 10.1016/j.numecd.2019.08.003; DOI: doi: 10.1039/d0fo00514b). In discussion, we have highlighted this limitation: “Although, the main limitation of the study is the lack of gut microbiota composition, we have previously documented an enhanced Lactobacillus counts in feces from rats treated with L. fermentum strain”.
- The authors should explain that why a mixed formulation with three L. fermentum strains were used in this study, rather than single strain or combined different species of Lactobacillus. Is there any data illustrate the effect of this three strains combined use is stronger than single strain?
Answer: We would like to thank the reviewer 2 for this suggestion. We rewrote the sentence to make the text clearer.
"Previously, the administration of L. fermentum 296 alone for four weeks was shown effective to reduce cholesterol and triglycerides serum levels in rats fed a HFD, but not to increase HDL-c levels and did not improve glucose tolerance and insulin sensitivity (Cavalcante, de Albuquerque et al. 2019). In this study, administration of mixed L. fermentum formulation decreased effectively the serum levels of glucose, cholesterol, LDL-c, triglycerides and atherogenic indices and increased HDL-c serum levels, demonstrating more effective hypolipidemic and hypoglycemic effects in rats fed a HFD”
- The author conducts animal experiments, but the data lacks tissue sections of intestinal or heart tissue, such as lesions, inflammation, and other related measurements.
Answer: We would like to thank the reviewer 2 for this comment. certainly, this will be carried in further studies of our laboratory.
Minor points:
- In Table 2, IL-1β and IL-6 belong to the inflammatory cytokine under the same inflammation pathway, and their levels in serum often show similar trends between groups. Since the administration of L. fermentum can significantly reduce the increase in IL-1β production induced by HFD, why this result is not seen in TNF-α and IL-6?
Answer: We would like to thank the reviewer 2 for their valuable comments. In fact, further studies will be need to know the signaling pathways by which L. fermentum formulation reduced IL1beta and increased IL-10, but did not change IL6 and TNFa in rats fed a HFD.
Although IL1beta and IL-6 belong to the inflammatory cytokine under the similar inflammation pathway, there is reports demonstrating that IL6 there is findings demonstrating pro- and anti-inflammatory properties of the IL-6 (doi:10.1016/j.bbamcr.2011.01.034). IL-1β belongs to the large family of interleukins 1 (IL-1), proinflammatory cytokines It is predominantly produced by monocytes and macrophages. IL-6 is produced by different tissues and cells, including endothelial cells, smooth muscle cells, monocytes, macrophages and adipose tissue, mainly the visceral, being released mainly by adipocytes. It can also act on different cell types, mainly with regard to immune and humoral effects related to inflammation. Once the inflammatory mechanisms are activated, IL-6 becomes crucial in its acute amplification, stimulating the production of acute phase proteins. TNF-α is released mainly by macrophages and lymphocytes. Knowing this, we recognized that more studies will be need to investigated the anti-inflammatory effect of L. fermentum in tissues (liver, gut and adipose tissue). Despite this, we findings demonstrated that, although subtle, L. fermentum attenuate inflammatory in rats fed a HFD.
- In Figure 1, since n = 6 in each group, why only showed 5 spots in some groups?
Answer: We would like to thank the reviewer 2 for this observation. during hplc experiments n=1 sample of control group and n=1 of HFD-LF were below of the limit detection. we have add this information in footnote of revised manuscript. as follows:
“During HPLC experiments acetate contents of 2 rats (01 of CTL group and 01 of HFD-Lf group) were below the analytical detection limit”
- In Figure 2, the authors should explain why fecal propionate contents were correlated negatively with serum levels of cholesterol, LDL-c, and triglycerides but did not correlate with HDL-c serum levels. Also, fecal propionate contents were correlated negatively with serum levels of TNF-α/IL-1β and correlated positively with serum levels of IL-6/IL-10 should be discussed.
Answer: We would like to thank the reviewer 2 for this comment. as suggested, in revised version we have included the discussion, as follow:
“In present study, administration of mixed L. fermentum formulation did not increase SCFA fecal contents in rats fed a HFD. However, fecal propionate contents were correlated nega-tively with serum levels of cholesterol, LDL-c, triglycerides TNFα and IL-1β, and correlat-ed positively with IL-10 and IL-6, indicating that dyslipidemia and pro-inflammatory conditions should be associated with reduced fecal propionate contents. These findings are consistent with data from early studies showing that fecal and circulating propionate exert inhibition effect on “de novo” lipogenesis and cholesterogenesis in the liver (Weitkunat, Schumann et al. 2016, Granado-Serrano, Martin-Gari et al. 2019). In addition, colonic propionate may inhibit NF-kB signaling with a consequent reduction of pro-inflammatory factors expression such as TNF-α and IL-1β in colon tissues (Filippone, Lanza et al. 2020), as well as, induce IL10-producing Treg cells (Smith, Howitt et al. 2013), being related as a novel candidate to improve metabolism and inflammation (Cani 2019)”.
- The sentence “ fermentum … was administered two twice a day ...” should be replaced to ““L. fermentum … was administered twice a day ...”” in Abstract section.
Answer: thank you. we have corrected.
- The terms of “ºC” in the Materials and Methods section should not underlined and should be replaced to “oC”.
Answer: thank you. we have corrected.
- The terms of “CFU/Ml” in the Materials and Methods section should be replaced to “CFU/mL”.
Answer: thank you. we have corrected.
- In the Materials and Methods section, “Homogenates were centrifuged at 1.180 g for 10 min”. Centrifugal speed should have type errors and should be corrected.
Answer: It was checked.
- In conclusion section, “…twice a day for four weeks” should be replaced to “…twice a week for four weeks”.
Answer: sorry, the correct is twice a day, for four week and not twice a week.
- When mention to bacteria, many species names are not italicized throughout the manuscript, please correct them.
Answer: thank you. we have corrected.
Reviewer 3 Report
Hello Authors,
This paper reviews the effect of mixed formulation of L. fermentum on cardiometabolic disorders. This formulation showed positive results in controlling adverse effects of high fat diets on some of the important health biomarkers. Here are some of the comments,
1) Include explanation of the gut-heart axis in the Introduction
2) What was the source of the strains obtained for this study? Plant, animal or something else?
Author Response
This paper reviews the effect of mixed formulation of L. fermentum on cardiometabolic disorders. This formulation showed positive results in controlling adverse effects of high fat diets on some of the important health biomarkers. Here are some of the comments,
We would like to thank the reviewer 3 for recognizing the relevance of our work. We appreciate the review of our manuscript and the great contributions that have been made. Below we answer all your questions.
1) Include explanation of the gut-heart axis in the Introduction
Answer: As suggested by reviewer 3, explanation on gut-heart axis was added in introduction section.
2) What was the source of the strains obtained for this study? Plant, animal or something else?
Answer: We would like to thank the reviewer 3 for this query. in revised version, we have added the source of the strains as follow:
“In last years, our research group have isolated and characterized potentially probiotic fruit-derived strains. The strains of L. fermentum 139, 263 and 296 were isolated of Brazilian fruit pulp processing by-products (Garcia, Luciano et al. 2016, de Albuquerque, Garcia et al. 2018). The strain L. fermentum 139 was isolated from the Mangifera indica L. (mango), L. fermentum 263 was isolated from the Ananas comosus and L. fermentum 296 was isolated from the Fragaria vesca L. (strawberry). All three strains displayed aptitudes as potential candidates for use as probiotics in a set of functionality-related in vitro properties, such as physiological functionalities of adhesion, aggregation, coaggregation, pathogen antagonism, and survival to the exposure to simulated gastrointestinal conditions, besides showing absence of hemolytic and mucolytic activities and resistance to antibiotics (de Albuquerque, Garcia et al. 2018). Such findings suggested for us that L. fermentum fruit-derived strains could be good candidates for inclusion in further in vivo studies to evaluate their beneficial health effects and confirm their potential for application as novel probiotics”.
Round 2
Reviewer 1 Report
The authors have made all the changes and corrections requested.
Author Response
We would like to thank the reviewer 1 again for recognizing the relevance of our work.
Reviewer 2 Report
The authors did not totally understand or respond to my questions.
- Since the authors keep emphasizing that probiotics decreases cardiometabolic disorders along with gut-heart axis, the authors should provide evidence about L. fermentum treatment improves cardiovascular disease via gut-heart axis, rather than just determined some inflammatory and oxidative stress factors, which could not represent gut-heart axis has been regulated. That means, the authors should do experiment to declare that the efficacy of these probiotics on diseases improvement is through gut-heart axis. Otherwise, the authors should not use the term of “gut-heart axis” in the title, abstract section, and whole manuscript.
- Gut microbiota should be analyzed to show the correlation between gut-heart axis and L. fermentum. The reference cited by the authors is not using the same probiotics formula for 16S sequencing at all.
- Also, English language and style should be edited and re-written in the whole manuscript since there are still many grammatical errors.
Author Response
Since the authors keep emphasizing that probiotics decreases cardiometabolic disorders along with gut-heart axis, the authors should provide evidence about L. fermentum treatment improves cardiovascular disease via gut-heart axis, rather than just determined some inflammatory and oxidative stress factors, which could not represent gut-heart axis has been regulated. That means, the authors should do experiment to declare that the efficacy of these probiotics on diseases improvement is through gut-heart axis. Otherwise, the authors should not use the term of “gut-heart axis” in the title, abstract section, and whole manuscript.
Answer: We would like to thank the reviewer 2 for all considerations and second opportunity to improve our manuscript. In fact, we agree with reviewer 2. In revised version, we re-write the title and we have removed the term “gut-heart axis” in whole manuscript. All correction in manuscript was given as yellow highlight text.
Gut microbiota should be analyzed to show the correlation between gut-heart axis and L. fermentum. The reference cited by the authors is not using the same probiotics formula for 16S sequencing at all.
Answer: we would like to thank reviewer 2 again. Unfortunately, we did not perform 16S sequencing in manuscript. In fact, we recognize the need of this experiment, but is not a routinely experiment realized in my Lab. There is outsourced service to perform the sequencing and bioinformatics analysis of gut microbiota. This will be performed in further studies of our laboratory. Despite this, we reiterate that gut microbiota analysis although is a limitation of our study, it does not invalidate the unpublished findings of the manuscript.
Sorry reviewer 2 by the cited reference mistake. In revised version, we have corrected the cited reference properly.
Also, English language and style should be edited and re-written in the whole manuscript since there are still many grammatical errors.
Answer: thank you reviewer 2. we have re-checked English language and corrected grammatical errors. in the whole manuscript.